# Deprivation-related and use-dependent plasticity go hand in hand

**Tamar R Makin[1]\*, Alona O Cramer[1], Jan Scholz[1,2], Avital Hahamy[3], David Henderson Slater[4], Irene Tracey[1,5], Heidi Johansen-Berg[1]**

[1]FMRIB Centre, Nuffield Department of Clinical Neuroscience, University of Oxford, Oxford, United Kingdom; [2]Mouse Imaging Centre, The Hospital for Sick Children, Toronto, Canada; [3]Department of Neurobiology, Weizmann Institute of Science, Rehovot, Israel; [4]Oxford Centre for Enablement, Nuffield Orthopaedic Centre, Oxford, United Kingdom; [5]Nuffield Division of Anaesthetics, University of Oxford, Oxford, United Kingdom

**Abstract** Arm-amputation involves two powerful drivers for brain plasticity—sensory deprivation and altered use. However, research has largely focused on sensory deprivation and maladaptive change. Here we show that adaptive patterns of limb usage after amputation drive cortical plasticity. We report that individuals with congenital or acquired limb-absence vary in whether they preferentially use their intact hand or residual arm in daily activities. Using fMRI, we show that the deprived sensorimotor cortex is employed by whichever limb individuals are over-using. Individuals from *either* group that rely more on their intact hands (and report less frequent residual arm usage) showed increased intact hand representation in the deprived cortex, and increased white matter fractional anisotropy underlying the deprived cortex, irrespective of the age at which deprivation occurred. Our results demonstrate how experience-driven plasticity in the human brain can transcend boundaries that have been thought to limit reorganisation after sensory deprivation in adults.

**\*For correspondence:** tamar.makin@ndcn.ox.ac.uk

**Competing interests:** The authors declare that no competing interests exist.

**Reviewing editor**: Ranulfo Romo, Universidad Nacional Autonoma de Mexico, Mexico

## Introduction

After losing a hand, simple tasks such as tying your shoelaces become a daily challenge, resulting in serious implications for quality of life and employment (*Jang et al., 2011*). How do different individuals adapt to such challenges, and what role does brain plasticity play in this adaptation? By addressing these questions in individuals with congenital or acquired hand absence we hope to shed light on the scope for adaptive plasticity in the adult human brain, thereby providing important information for informing future neurorehabilitation strategies.

It has been thought that adaptive plasticity is restricted in the adult primary sensory cortex following sensory deprivation, as shown recently in late blindness (*Baseler et al., 2011*), suggesting limited scope for neurorehabilitation following sensory deprivation. Arm-amputation is a particularly powerful model for studying plasticity as it combines two major drivers for reorganisation – sensory deprivation and adaptive motor behaviour. Despite this, most accounts of plasticity following arm amputation focused on sensory deprivation, and in particular on apparently passive remapping of adjacent face or arm representation into the deprived cortex (*Lotze et al., 1999*; *Ramachandran and Altschuler, 2009*; *Nava and Röder, 2011*). Although longer-range remapping has also been reported, such as intact hand representation in the deprived cortex (*Bogdanov et al., 2012*), this is usually also explained as a passive result of inter-hemispheric dis-inhibition (*Werhahn et al., 2002*; *Ramachandran and Altschuler, 2009*; *Simões et al., 2012*). However, these accounts ignore substantial adaptations in motor behaviour that accompany absence or loss of a limb, which could be powerful drivers for plasticity (*Scholz et al., 2009*).

**eLife digest** The loss of a limb will have a profound impact on an individual's daily life. Nevertheless, individuals can employ a variety of behavioural strategies to adapt to the loss of, say, a hand. Some become skilled at using the residual part of their arm, while others prefer to rely on their other hand. Their brain, too, will undergo major changes. Many studies have shown that the region of the brain that controlled a given limb can be "taken over" by another part of the body if that limb is lost. This process has been previously considered to be harmful, as it has been linked to experiences of pain arising from the missing limb.

Now, Makin et al. have explored the links between changes in the behaviour of individuals missing a hand and changes in their brains. People who had been born without a hand or who had lost a hand in later life were asked to wear a device that recorded their movements as they went about their daily lives. The data revealed that people who had been born without a hand made relatively more use of their residual limb, while those who had lost their hand made relatively more use of their remaining hand.

Moreover, these differences were reflected in patterns of brain activity. In the subjects born without a hand (who were making relatively extensive use of their residual limb), the area of the brain that would otherwise control the 'missing' hand was activated when the subjects moved their residual limb. And in the subjects who had lost their hand, this brain region was activated when they moved their remaining hand. However, in individual subjects, the size of the effect depended on the usage preferences of the subject: for example, the minority of people who were born without a hand but nevertheless make extensive use of their intact hand showed a pattern of activation that resembled the average pattern seen in those who had lost a hand in later life.

By providing new insights into the plasticity of brain and behaviour following the loss of a hand, the work of Makin et al. may aid the development of rehabilitation techniques to help patients to optimise the use of both their residual and their intact limbs.

Here, we test whether altered limb–use patterns influence cortical reorganisation in individuals with unilateral hand absence, using functional magnetic resonance imaging (fMRI) and diffusion tensor imaging (DTI). One aim of the current study was to assess upper limb use strategies in individuals with acquired and congenital hand absence. A further aim was to investigate whether functional cortical reorganisation underlies any observed variation in motor behaviour. Specifically, we predicted that the degree to which individuals used their residual-arm or intact-hand in daily life would reflect the degree of brain activity in the deprived cortex during movement of the corresponding limb.

## Results

### Assessing limb use strategies in individuals with acquired and congenital hand absence

We assessed day-to-day limb-use strategies in individuals with acquired (n = 18, 10 below elbow) or congenital (n = 11, 10 below elbow) hand absence (see *Table 1* for demographic details and prosthetic limb usage). Specifically, we quantified the extent to which individuals utilise their residual arms, relative to their intact hands (*Figure 1A*), in daily activities. We collected data from acceleration monitors, worn by a sub-set of 20 participants (8 congenital) while engaged in their normal routines. A laterality index, reflecting the relative number of movements performed by the intact hand vs the residual arm (*Figure 1B*), showed that both groups moved the intact hand more than the residual arm (t(7) = 3.48, p=0.01; t(11) = 10.60, p=0.001, one-sample t-test compared to zero for the congenital and acquired groups, respectively). This could be due both to greater use of the intact hand and also to the different position of the accelerometer on the two sides of the body (wrist vs arm), resulting in different acceleration profiles. Our main aim was to compare the degree of this laterality across the two 1-handed groups. We found that the acquired group exhibited stronger relative reliance on their intact hand, as demonstrated in a significantly greater lateralisation scores towards the intact hand, compared to the congenital group (t(18) = −2.67, p=0.016, *Figure 1B*).

**Table 1.** Demographic details of 1-handed individuals with acquired (A) and congenital (C) hand loss

|  | Age | Deprivation age (in years) | Amp. Level | Side/dominant | Cause of amputation | Cosmetic Pros. Usage | Functional Pros. Usage |
|---|---|---|---|---|---|---|---|
| A01 | 43 | 38 | 4 | L/R | Trauma | 2 | 0 |
| A02 | 42 | 22 | 4 | R/L | Nerve I* | 2 | 0 |
| A03 | 21 | 18 | 4 | R/L | Trauma | 0 | 0 |
| A04 | 46 | 37 | 2 | L/R | Nerve I* | 1 | 0 |
| A05 | 48 | 20 | 1 | R/R | Trauma | 1 | 5 |
| A06 | 58 | 11 | 2 | R/R | Trauma | 1 | 5 |
| A07 | 31 | 2 | 2 | L/R | Trauma | 0 | 0 |
| A08 | 54 | 20 | 5 | L/L | Trauma | 5 | 0 |
| A09 | 47 | 45 | 2 | L/L | Tumour | 1 | 3 |
| A10 | 60 | 34 | 2 | R/R | Trauma | 0 | 5 |
| A11 | 51 | 35 | 4 | L/R | Infection | 1 | 5 |
| A12 | 47 | 19 | 2 | L/R | Trauma* | 0 | 5 |
| A13 | 57 | 48 | 4 | R/L | Infection | 0 | 2 |
| A14 | 56 | 40 | 2 | L/R | Trauma | 0 | 0 |
| A15 | 22 | 18 | 5 | L/R | Trauma | 0 | 0 |
| A16 | 43 | 33 | 4 | L/R | Trauma | 0 | 5 |
| A17 | 50 | 28 | 4 | L/R | Trauma | 5 | 0 |
| A18 | 52 | 45 | 4 | L/R | Trauma | 2 | 5 |
| C01 | 31 | 0 | 4 | R | Dysmelia | 5 | 0 |
| C02 | 24 | 0 | 4 | L | Dysmelia | 4 | 0 |
| C03 | 35 | 0 | 4 | L | Dysmelia | 5 | 0 |
| C04 | 31 | 0 | 5 | L | Dysmelia | 0 | 0 |
| C05 | 25 | 0 | 4 | L | Dysmelia | 0 | 0 |
| C06 | 54 | 0 | 4 | L | Dysmelia | 0 | 5 |
| C07 | 49 | 0 | 5 | L | Dysmelia | 0 | 0 |
| C08 | 22 | 0 | 4 | R | Dysmelia | 1 | 0 |
| C09 | 49 | 0 | 4 | R | Dysmelia | 4 | 0 |
| C10 | 18 | 0 | 4 | L | Dysmelia | 0 | 0 |
| C11 | 46 | 0 | 2 | L | Dysmelia | 2 | 5 |

Amputation levels: the level at which the residual arm ends. 1 = through shoulder, 2 = above elbow, 3 = through elbow, 4 = below elbow, 5 = through wrist; Side = side of amputation; dominant = hand dominance prior to hand loss (based on self report): L = left, R = right; NI=nerve injury.
*indicates potential partial spinal damage. Pros. Usage = Prosthetic limb usage (frequency): 0 = never, 1 = rarely, 2 = occasionally, 3 = daily, 4 = more than 4 hr a day, 5 = more than 8 hr a day.

These findings were further confirmed using questionnaire ratings from all 1-handed participants, quantifying the extent to which individuals incorporate their residual arm in daily tasks ('Materials and methods'). Ratings were significantly greater for the congenital group (t(1,27) = 3.65, p=0.001), suggesting they use their residual arm more frequently in their daily routine, compared with the acquired group (*Figure 1C*). This group difference remained significant after accounting for level of amputation and degree of functional and cosmetic prosthetic limb usage as covariates ($F_{(1,24)}$ = 8.217, p=0.009). The questionnaire ratings for residual arm usage were negatively correlated with the accereometry-based laterality indices ($r_{(18)}$ = −0.53, p=0.012, *Figure 1D*), validating the questionnaires as a measurement of habitual usage strategies between the residual and intact limbs. A similar correlation was also obtained after accounting for level of amputation and degree of functional and cosmetic prostheses usage ($r_{(15)}$ = −0.451, p=0.034).

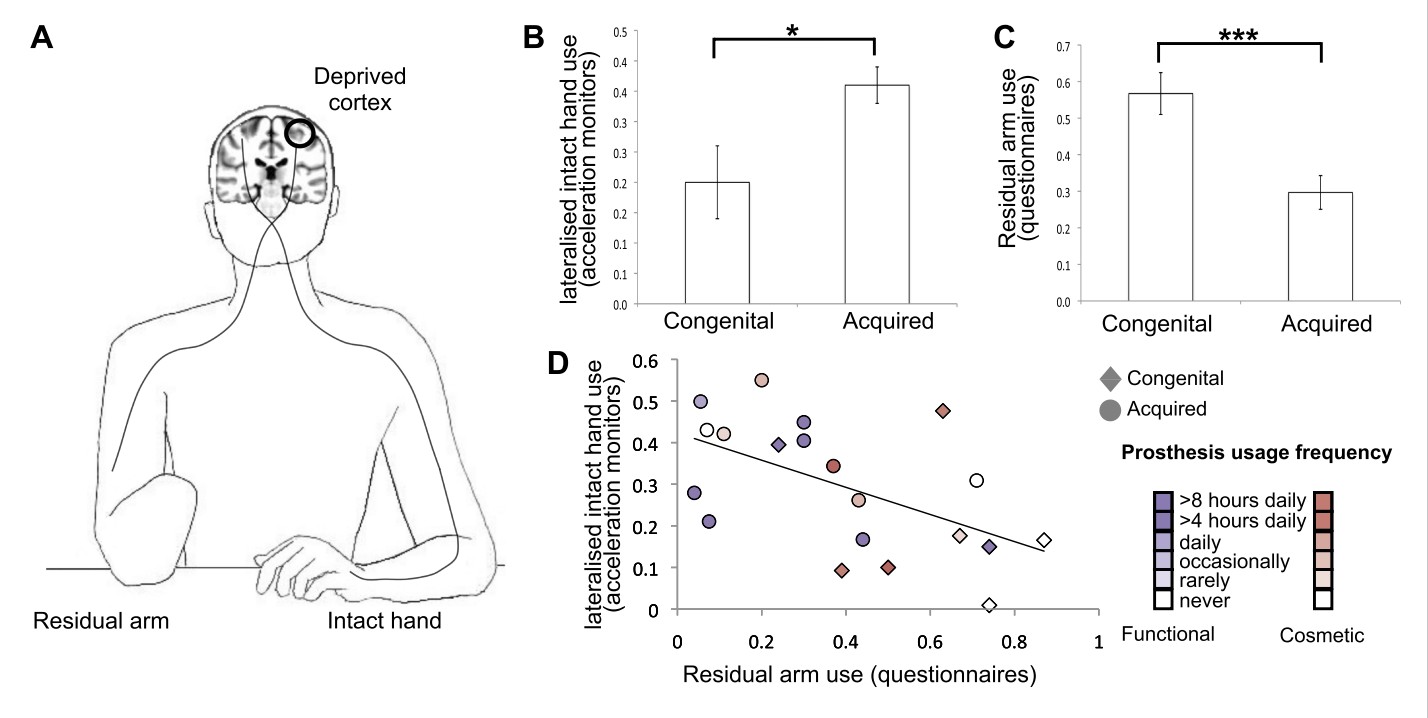

**Figure 1**. Dissociative limb usage strategies in 1-handed individuals with congenital and acquired hand loss. (**A**) Schematic illustration of the residual and intact limbs. (**B**) Limb-use strategies, based on activity monitors for increased lateralised (intact hand) use. A laterality index, reflecting the relative number of movements performed by the intact hand vs the residual limb, was calculated using data from acceleration monitors (mean ± s.e.m.), worn by 20 1-handed individuals (8 congenital) while engaged in their normal routines. Positive values represent a tendency to use the intact hand more than the residual limb. Although both groups exhibit such a tendency (t(7) = 3.48, p=0.01; t(11) = 10.60, p=0.001, one-sample t-test compared to zero for the congenital and acquired groups, respectively), the acquired group shows a significantly greater preference for the intact hand compared to the congenital group (t(18) = −2.67, p=0.016). (**C**) Questionnaire ratings (mean ± s.e.m.) for residual arm usage in daily activities were significantly greater for the congenital group (t(1,27) = 3.65, p=0.001), suggesting they use their residual arm more frequently in daily activities, compared with the acquired group. (**D**) Questionnaire ratings for residual arm usage were negatively correlated with the laterality indices, measured based on acceleration monitoring (r(20) = −0.53, p=0.012), validating the questionnaires as a measurement of habitual usage strategies between the residual and intact limbs. Scatter plot shows data for 1-handed individuals with congenital and acquired hand loss, frequency of prosthetic hand usage is indicated in the index to the right. Asterisks denote significance levels of *p<0.05; ***p<0.005.

## Whole brain comparisons of limb use representation in individuals with acquired and congenital hand absence

Based on these dissociated adaptive motor strategies, we predicted distinct patterns of limb representation between the two 1-handed groups. All 1-handed participants and 22 intact (2-handed) controls underwent fMRI, involving simple unilateral hand and arm movements (fingers/elbow flexion and extension). Voxel-wise activation (β) patterns were compared between the three groups during hand and arm movements across the entire brain. Each of these two separate whole-brain contrasts resulted in a single cluster, centred on the hand knob of the central sulcus of the deprived hemisphere, spanning both the pre- and the post-central gyri. In line with their increased usage of the residual arm, congenital 1-handed participants showed increased activation during residual arm movements, compared with the other two groups (*Figure 2B*). The resulting cluster showed two distinct local peaks of activation—in the anterior and posterior banks of the central sulcus (t = 3.73; x = 38, y = −20, z = 54 for the anterior peak and t = 3.93; x = 48, y = −20, z = 54 for the posterior peak, coordinates are based on the MNI 152 brain template). Conversely, acquired 1-handed participants, who displayed increased use of their intact hand, correspondingly showed significantly increased activation during intact hand movements compared to the other groups (*Figure 2C*). The resulting cluster had a single peak, centred on the posterior bank of the central sulcus (t = 5.24; x = 42, y = −18, z = 48 in MNI 152 space).

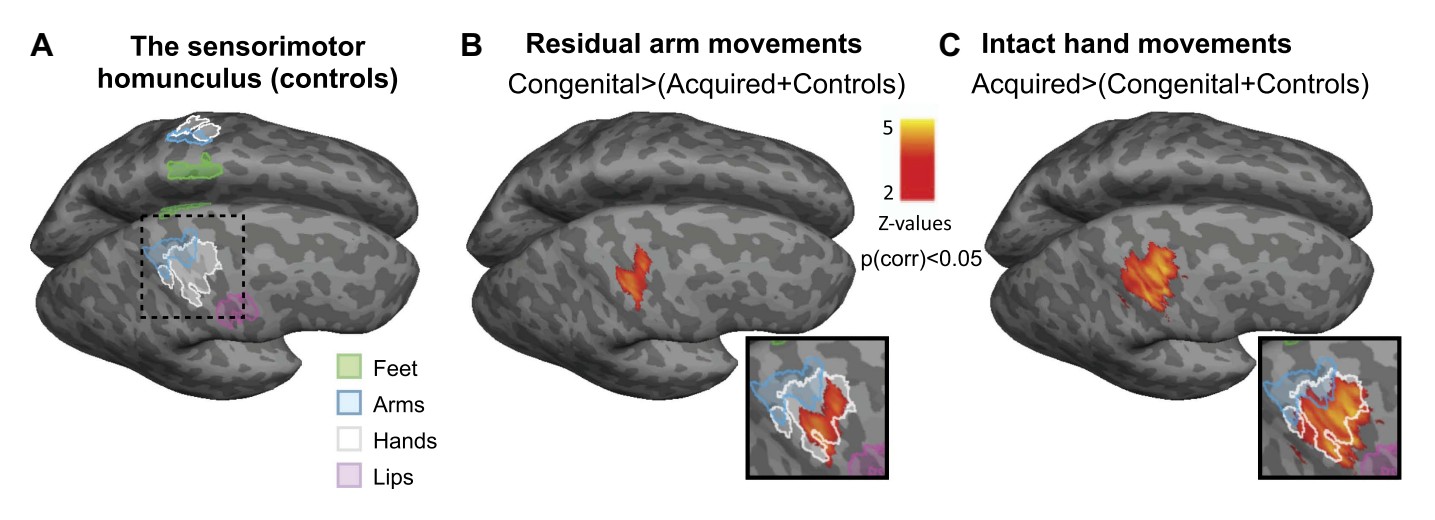

**Figure 2**. Limb-representation patterns in the deprived cortex reflect usage—whole brain contrasts. (**A**) Coloured lines delineate the boundaries of clusters activated during execution of movements using the feet (green), arms (blue), hands (white), and lips (pink) in controls, projected on inflated hemispheres. (**B**) Whole-brain group comparisons for activation during residual/nondominant arm movements (in 1-handed/control participants, respectively; deprived hemisphere is in front. Note that participants with above elbow deprivation were excluded). During movements of the residual arm, the congenital group showed increased activation compared with the acquired and control groups. This whole-brain contrast resulted in a single cluster, centred on the hand knob of the central sulcus of the deprived cortex, spanning the pre- and post-central gyri (shown in orange). (**C**) Whole-brain group comparisons for activation during intact/dominant hand movements (in 1-handed/control participants, respectively). During movements of the intact hand, the acquired group showed increased activation compared with the conjunction of the congenital and control groups. This whole-brain contrast resulted in a single cluster centred on the hand knob of the deprived cortex spanning the pre- and post-central gyri (shown in orange). The square inserts in (**B** and **C**) show overlap between the clusters resulting from the whole brain group comparisons (orange) and the controls' hand area (white), as shown in (**A**). No other significant clusters were identified here, or using the homologous contrasts between the other groups.

Even though activations used to generate these two contrasts were measured during movements of different limbs (hand vs arm), and on different sides of the body (limbless side vs intact side), the differential activation was restricted in both cases to the hand area, as demonstrated by activation patterns in the control group (*Figure 2A* and inserts in *Figure 2B,C*). No other significant clusters were identified in these comparisons, or using the homologous contrasts between the other groups.

## Limb use representation patterns in the 'deprived cortex'

As demonstrated in *Figure 2*, the two independent clusters, showing increased representation of the residual arm or the intact hand in the congenital and acquired groups, overlapped with the hand area of the control participants. This suggests that the area that would typically represent the missing hand has been recruited to support increased representation of the residual arm or intact hand. To investigate use-dependent plasticity specifically in the 'deprived cortex', we constructed an independent region of interest (ROI), based on the conjunction between phantom/nondominant hand movements in acquired amputees/2-handed controls. Amputees often experience vivid sensations of a phantom hand. Recently, this phenomenology has been supported by empirical evidence demonstrating that movements of a phantom limb elicit both central and peripheral motor signals that are different from those found during imagined movement (*Reilly et al., 2006*; *Raffin et al., 2012*). This phenomenon of maintained representation in the sensorimotor cortex, allowed us to reliably localise the representation of the missing hand (*Makin et al., 2013*). Here, this pre-defined ROI was used to interrogate activity relating to other body parts, thus allowing us to test the usage-driven hypotheses independently. The borders of this 'deprived cortex' ROI, spanning primary somatosensory and motor cortices, are shown in *Figure 3A*. As the spatial resolution and co-registration methods used here are insufficient to reliably dissociate the somatosensory and motor primary cortices, cortical areas spanning the pre- and post-central gyri will be described here as 'sensorimotor'.

We next extracted fMRI activation values (β) during movement conditions within the independent ROI. A mixed effect analysis of variance showed a significant interaction between groups and limbs ($F_{(2,47)} = 3.67$, p=0.033), reflecting dissociated recruitment of the deprived cortex by movements of the

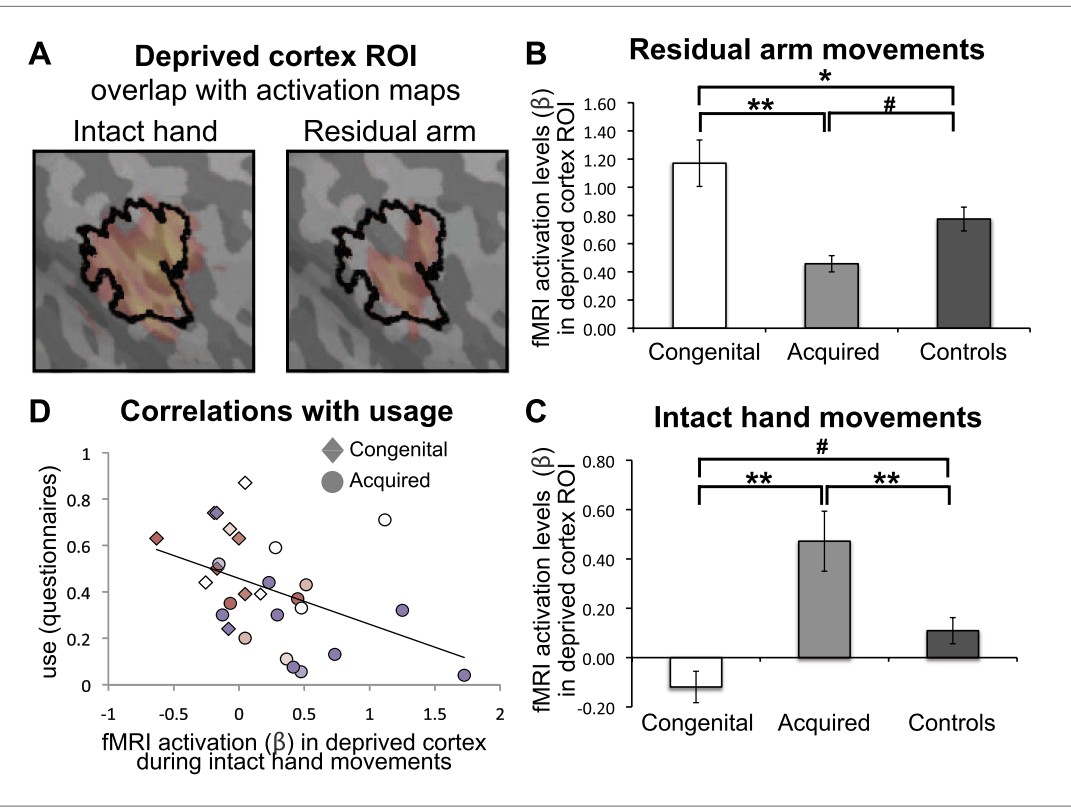

**Figure 3**. Limb-representation patterns in the deprived cortex reflect usage—ROI analysis. (**A**) To define the deprived cortex, an independent ROI (outline shown in black) was derived from the conjunction between phantom/nondominant hand movements in acquired amputees/controls. For illustration purposes, the group contrast maps derived from the previous whole-brain analysis (shown in *Figure 2*) are also overlaid (in faded orange), highlighting the fact that whole-brain group differences are co-localised with the deprived cortex ROI. (**B** and **C**) In order to assess the degree to which the deprived cortex is used to represent other body parts, mean fMRI activation levels ($\beta$) for voxels within the deprived cortex ROI during residual arm (**B**) and intact hand (**C**) movements were calculated. Beta values were averaged (±s.e.m) across the congenital (white), acquired (light gray) and control (black) groups. Within-group comparisons confirmed greater representation of the relatively over-used limb in each group. Asterisks/Hashes denote significance levels of *$p \leq 0.025$; **$p < 0.005$; #$p < 0.05$ for corrected (planned)/and uncorrected (exploratory) comparisons, respectively. (**D**) Increased fMRI activation ($\beta$) in the deprived cortex during intact hand movements correlated negatively with questionnaire scores for residual arm usage (associated with increased intact hand usage, *Figure 1D*), across the congenital and acquired groups (r(28) = −0.43, p=0.021). Prostheses usage indices are shown in *Figure 1*.

residual arm and intact hand in the two 1-handed groups (*Figure 3B–C*). Post-hoc tests confirmed increased activation when congenital participants moved their (relatively over-used) residual limb, compared to acquired participants (Mann Whitney U = 6, n = 20, p=0.001) and compared to controls (t(30) = 2.37, p=0.025). Similarly, greater activation within the deprived cortex was observed when acquired participants moved their (relatively over-used) intact hand, compared to congenital participants (t(26) = −3.68, p=0.001) and controls (t(37) = 2.96, p=0.005). These results were independent of participant A07, who lost her hand at a young age (t>2.81, p<0.008) (note that this participant was not included in the arm analysis above, due to her amputation level). The difference between the two 1-handed groups was also significant when accounting for level of amputation, and degree of functional and cosmetic prosthetic limb usage ($F_{(1,15)}$ = 11.357, p=0.004 and $F_{(1,23)}$ = 8.172, p=0.009 for residual arm and intact hand movements, respectively).

To complement the between-group findings reported above, we also performed within-group analyses to test for assymetrical patterns of limb representation. In particular, we tested whether the activity in the deprived cortex during movement of the over-used (compensatory) limb was greater than activity in the homologous cortex (in the intact hemisphere) during movements of the corresponding (opposite) limb (*Figure 4*). For the congenital group, activation in the deprived cortex during movements

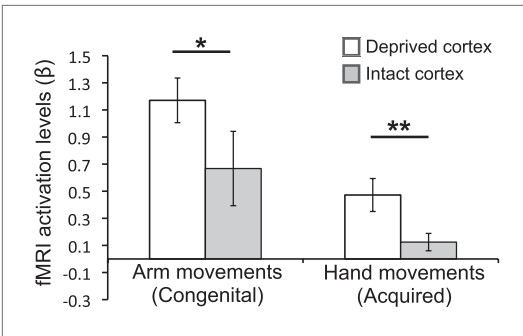

**Figure 4**. Over-representation of the favoured limb for adaptive use results in intra-subject assymetry. Activation levels (mean ± s.e.m) during arm movements in the congenital group and during hand movements in the deprived cortex ROI (white), and a homologous ROI contralateral to the intact hand (gray). The congenital group showed increased contralateral activation (in the deprived cortex) during residual arm movements, compared to contralateral activation (in the homolgous intact cortex) during intact arm movements. The acquired group demonstrated significantly increased ipsilateral activation (in the deprived cortex) when moving their intact hand, compared with ipsilateral activation (in the intact cortex) during phantom hand movements. These results confirm adaptive changes in limb-representation in the deprived cortex in 1-handed individuals. Asterisks denote significance levels of *p<0.05; **p<0.01.

of the (contralateral) residual arm was greater than activation in the homolgous intact cortex during movements of the (contralateral) intact arm (t(1,9) = 2.45, p=0.037). Similarly, for the acquired group, activation in the deprived cortex during movements of the (ipsilateral) intact hand was greater than activation in the homolgous intact cortex during movements of the (ipsilateral) phantom hand (*Makin et al., 2013*) (t(1,16) = 3.08, p=0.007). These results confirm that, within both groups, activity in the deprived cortex relating to the compensatory limb is greater than activity in the homologous cortex, relating to the corresponding opposite limb.

In addition to the hypothesised increase in activation for the limb favoured for adaptive usage, the ROI analysis in *Figure 3B–C* also revealed unexpected reduced activation in the 1-handed groups during movements of the limb which is not favoured for over-use, compared with controls. Specifically, the congenital group, who over-use the residual arm, showed reduced activation in the deprived cortex during movements of the intact hand (t(31) = −2.61, p=0.014), whereas the acquired group, who over-use the intact hand, showed reduced activation in the deprived cortex during movements of the residual arm (Mann Whitney U = 97, n = 39, p=0.025). This result potentially reflects a competitive relationship between the representations of the two limbs in the deprived cortex in 1-handed individuals.

To test whether functional reorganisation associated with adaptive movement strategies is present outside the primary sensorimotor cortex, we also explored patterns of limb representation in the cerebellar hand area (lobule V of the anterior lobe) of the acquired and congenital 1-handed groups. We defined an independent ROI for the deprived cerebellum, using the same criteria as used to construct the deprived cortex ROI. Note that the deprived cerebellum ROI is ipsilateral to the missing hand (and opposite to the deprived cortex ROI). Similarly to the deprived cortex, the deprived cerebellum showed increased activation in the congenital group during residual arm movements (t(26) = −3.33, p=0.003), and increased activation in the acquired group during intact hand movement (t(18) = 2.75, p=0.013). This suggests that the patterns of altered representation identified in this study are not limited to reorganisation of body-part representations in the primary sensorimotor cortex.

## Accounting for potential confounds of handedness and phantom pain

For the group results described above, we have compared the intact hand of 1-handed participants to the dominant hand of controls, and the residual arm of 1-handed participants to the non-dominant arm of controls. However, a third of the individuals with acquired deprivation lost their dominant hand (*Table 1*). To test whether group differences in hand dominance could influence the results, activation levels for non-dominant hand and dominant arm movements were extracted from a homologous ROI contralateral to the dominant hand, in an age-matched sub-group of the control participants. ROI-based comparisons between the acquired and control groups were repeated as above, while accounting for handedness using this sub-group, with similar results (t(30) = 3.1, p=0.004 for increased representation during intact hand movements and t(37) = −2.25, p=0.032 for decreased residual arm representation during residual arm movements in acquired amputees vs controls).

Phantom limb pain, prevalent in acquired amputees, may discourage these participants from using their residual arm. Indeed, within the acquired group a trend for a correlation between chronic

phantom pain scores and usage was found ($r_{(16)}$ = −0.40, p=0.099), such that individuals with worst chronic pain tend to use their residual arm less. Therefore, to test whether phantom pain could explain our observed imaging results, we repeated the comparisons between the two 1-handed groups described above (using a univariate GLM) with chronic phantom pain ratings as a covariate. Increased activation was maintained both in the congenital group (relative to the acquired group) during residual arm movements ($F_{(1,19)}$ = 12.57, p=0.002) and in the acquired group (relative to the congenital group) during intact hand movements ($F_{(1,27)}$ = 6.10, p=0.021). These results suggest that the activation patterns presented above are not merely epiphenomenal to phantom pain.

## Over-representation of the intact hand in the deprived cortex correlates with adaptive limb use

Although the congenital and acquired 1-handed groups differed in terms of usage strategies and activation patterns in the deprived cortex, individuals across these groups showed considerable overlap (*Figure 3D*): acquired 1-handed individuals who nevertheless had high residual arm usage showed less of the intact hand activation that was typical of their group. Such individuals demonstrate that the differential reorganisational patterns observed in the two 1-handed groups may not be solely determined by deprivation age (or previous afferent/efferent experience), but rather could relate to usage strategies. Indeed, when considering usage across *both* 1-handed groups we found that greater usage of the residual arm was associated with less activation during intact hand movements in the deprived cortex ($r_{(26)}$ = −0.43, p=0.021), even when accounting for age of sensory deprivation ($r_{(25)}$ = −0.36, p=0.032). This effect was strengthened when repeating the same partial correlation while only considering the prosthetic limb users ($r_{(19)}$ = −0.61, p=0.003 and $r_{(18)}$ = −0.56, p=0.005, respectively), suggesting that rehabilitation may strengthen the relationship between adaptive plasticity and usage, however further research will be required to identify the relationship between prosthesis usage and adaptive brain plasticity.

## Over-representation of the intact hand in the deprived cortex correlates with increased white-matter fractional anisotropy

Finally, the white matter connections supporting the functional changes associated with limb-usage were studied in voxel-wise comparisons (*Smith et al., 2006*) of fractional anisotropy (FA, a measure of white matter microstructure), along the white matter skeleton. No significant differences in FA were found between the three groups, suggesting that deprivation in itself may be insufficient to induce significant white matter change (although see *Langer et al. (2012)* for a study showing reduced FA in the corticospinal tract following several weeks of arm immobilisation). However, across both 1-handed groups, individuals showing greater activation in the deprived cortex during intact hand movements had higher FA values in the corticospinal tract and the inferior fronto-occipital fasciculus (IFO) in the deprived hemisphere (*Figure 5*). This voxel-wise non-parametric correlation was performed while accounting for participants' age and deprivation age (as nuisance regressors). No significant correlations were identified between FA and degree of activation in the deprived cortex during residual arm movements.

## Discussion

We provide the first evidence that altered patterns of adaptive limb use, in individuals with unilateral hand absence, are reflected in distinct patterns of cortical reorganization. Using physiological measurements, we provide the first detailed description of patterns of daily limb usage in individuals with congenital and acquired limb-loss. Next, using fMRI we show that representation patterns in the deprived cortex of 1-handed individuals are contingent upon the limb-use strategy adopted by individuals, rather than a sensitive period in development: Individuals from either 1-handed group that rely more on their intact hands showed increased representation of the (ipsilateral) intact hand in the deprived cortex, irrespective of age at deprivation. These functional adaptations are further reflected in white matter structural integrity, as demonstrated using DTI. These findings shed new light on the extent of adult brain plasticity and have implications for rehabilitation.

Loss of a limb can have devastating impact on quality of life in individuals who are typically young, employed and otherwise healthy. As a result of limb loss, 69% of sufferers have to change job or become unemployed (*Jang et al., 2011*). How best to rehabilitate these individuals, and allow them to return to active and productive lifestyles, is unknown (*Nimhurchadha et al., 2013*). While tremendous

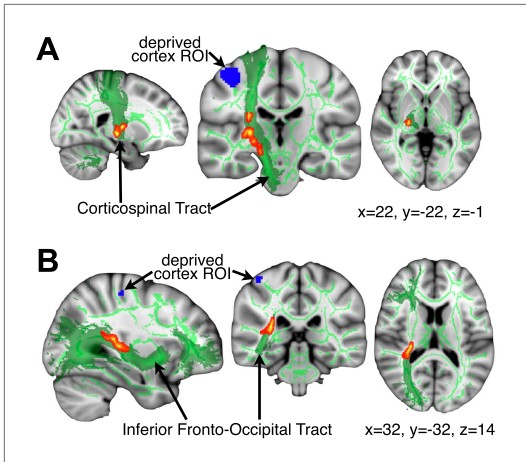

**Figure 5**. Higher FA is associated with greater intact hand plasticity in 1-handed participants. Clusters in the corticospinal tract (top) and the inferior fronto-occipital fasciculus (bottom) (red-yellow, p<0.05, corrected) show positive correlations between FA and intact hand fMRI activation within the deprived cortex ROI (blue), while accounting for participants' age and deprivation age. The bright/dark green lines denote the white matter skeleton/tracts (based on the John Hopkins University atlas), respectively. Clusters have been dilated for presentation purposes.

resources are dedicated to aiding 1-handed populations through the development of sophisticated prosthetic limbs (*Kuiken et al., 2009*), a relatively large number of the individuals do not use them and prefer to carry out daily activities with their intact hand (*Jang et al., 2011*; *Østlie et al., 2012*). Clinicians are aware that different individuals adapt to limblessness in varying ways–with some relying on the intact limb (*Jones and Davidson, 1999*) and others showing remarkable ability with their residual limb or a prosthetic limb (*Smurr et al., 2008*; *Jang et al., 2011*). Surprisingly, however, to our knowledge there has been no previous empirical study of the range of adaptive motor behaviours taken in 1-handed individuals in their natural environment (*Lindner et al., 2010*). Based on subjective (questionnaires) and objective (accelerometry) usage monitoring, we provide empirical data on patterns of limb usage in these groups while engaged in their natural routine (*Figure 1*) and then show that these daily usage strategies are a powerful driver of plasticity in the deprived cortex.

Development of compensatory skills, whether through rehabilitation or through the natural course of adaptation, critically depends on learning and brain plasticity. Previous studies of maladaptive brain changes after amputation (*Flor et al., 1995*) have been instrumental in guiding new evidence-based rehabilitation interventions to treat phantom pain (*Chan et al., 2007*; *Lotze et al., 1999*). However, the study of brain changes after amputation has not previously been brought to bear on the clinical problem of adaptive motor behaviour. Understanding the scope for adult human brain plasticity is a key consideration for designing successful future neurorehabilitation approaches (*Krakauer, 2006*; *Koenig et al., 2013*).

We found that patterns of increased representation of displaced body parts in the deprived cortex closely relate to adaptive daily strategies of limb-usage in individuals with upper limb absence: people with congenital hand absence, who are better at incorporating their residual arms in daily tasks, activate their missing hand cortex during residual arm movements (*Figures 2B, 3B, 4*). This is consistent with previous reports of reorganisation associated with compensatory foot-use in individuals with bilateral congenital upper limb malformation (*Stoeckel et al., 2009*). Conversely, the 1-handed individuals with acquired amputation, who are more dependent on their intact hands for daily activities, show strong activation in their deprived cortex when the intact hand is moving (*Figures 2C, 3C, 4*). However, this dissociative representation doesn't exclusively rely on the cause of or age at deprivation: increased ipsilateral intact hand representation was associated with the usage strategy adopted by individuals (*Figure 3D*), such that people who have learned to incorporate their residual arms in daily tasks tended to show reduced intact hand representation in the deprived cortex.

Using both whole-brain and ROI analyses, our functional results demonstrate how plasticity within the deprived cortex may be shaped by adaptive experience. However, it is possible that adaptive strategies for compensatory limb-usage drive similar, yet subtler differences in other brain areas underlying limb usage. For example, using an ROI analysis, we have identified similar patterns of limb over-representation in the deprived cerebellum, which were not apparent in the (corrected) whole-brain comparison. We therefore cannot rule out the possibility that habitual usage strategies could produce further changes in brain representation, beyond the deprived cortex and even the sensorimotor system. Further research is therefore necessary in order to determine the full extent of adaptive plasticity following limb-loss.

Finally, while no group differences in FA values were identified, the degree of intact hand representation in the deprived cortex correlated with FA values along the IFO and the corticospinal tracts of

the deprived cortex (*Figure 5*). The IFO has been recently implicated in visuospatial representation, as well as visuomotor control (*Urbanski et al., 2008*; *Migliaccio et al., 2012*), which may be relevant for lateralised limb usage (*Makin et al., 2010*). The structural differences in the corticospinal tract, associated with increased representation of the intact hand in the deprived cortex, may shed new light on the origin of bimanual plasticity in amputees, which has previously been regarded as a consequence of reduced inter-hemispheric inhibition (*Simões et al., 2012*). Indeed, we find similar results in the deprived cerebellum, as identified in the sensorimotor cortex, despite fundamental differences in body-part topography (*Shambes et al., 1978*) in these two brain regions. One tentative interpretation of these findings is that longer-range plasticity (where the deprived cortex is recruited by the intact hand) elicits structural white matter change, whereas more local plasticity (where the deprived cortex is recruited by the residual arm) does not. However, the cross-sectional nature of the study does not allow us to differentiate between experience-dependent and pre-existing variations in white matter structure that might confer a bias towards intact hand representation in the deprived cortex.

To conclude, we demonstrate how adaptive limb usage strategies may serve as powerful drivers of both functional and structural plasticity in adults. We show that the deprived cortex in people with either congenital or acquired hand absence is employed by whichever part of the upper limb individuals are relatively over-using (compared to other 1-handed individuals) to compensate for their disability. This occurs irrespective of whether this is a hand or an arm and the degree to which it is typically represented in the deprived brain area. By demonstrating that adaptive plasticity in amputees transcends the boundaries thought to restrict reorganisation after sensory deprivation in the adult human brain, our results may inspire future research, aimed at utilising neurorehabilitation to improve usage of both residual and intact arms, as well as artificial arms, in 1-handed individuals of all ages.

## Materials and methods

### Participants

18 individuals with sustained unilateral upper limb amputation (mean age ± s.e.m = 46 ± 3, 6 with absent right hand), 11 individuals with a congenital unilateral upper limb deficit (age = 35 ± 4, 3 with absent right hand) and 22 intact controls (age = 41 ± 3, 7 left hand dominant) were recruited for the study (see *Table 1* for demographic details). Recruitment was carried through the Oxford Centre for Enablement and Opcare in accordance with NHS national research ethics service approval. Informed consent and consent to publish was obtained in accordance with ethical standards set out by the Declaration of Helsinki (1964) and with procedures approved by the NHS (REC ref: 10/H0707/29). Data from one individual with acquired deprivation and one control were discarded, due to excessive head movements (>3 mm) during fMRI and problems during DTI data acquisition (respectively).

### Limb-usage strategy measurements

In order to assess potential differences in rehabilitation strategies between the two amputee groups, use of residual arm was initially assessed using a revised version of the Motor Activity Log (*Uswatte et al., 2006*). 1-handed participants were requested to rate how frequently (never; sometimes; very often) they incorporate their residual arm (stump; either directly, or using a prosthesis) in an inventory of daily activities, requiring varying degrees of motor control. The following items were used: taking money out of wallet; opening envelope; picking up/lifting glasses; picking up/holding up a phone; wiping off a kitchen counter or other surface; getting out of a car; stabilizing paper while writing; stabilizing dishes while carrying; carrying a cup or a can; carrying bags; getting up from a chair with arm rests; pulling chair away from table before sitting down; holding a book or a magazine/turning pages; typing on a keyboard/pressing mouse buttons; controlling a computer mouse; putting on your socks; putting on your shoes; tying shoe laces; inserting your (intact) arm through a sleeve; putting on makeup base, lotion, or shaving cream on face; washing hand or face; drying your hand or face; combing your hair; buttoning a shirt; zipping up a coat; peeling fruit skin; using a fork or spoon for eating.

As previous motor activity logs were designed to assess mobility of a paralyzed hand, the questionnaire was modified while considering the unique confounds of unilateral amputees (*Lindner et al., 2010*), and was aimed to accommodate participants with various levels of deprivation. In accordance

with the original questionnaire, the items represented commonly encountered actions and covered a comprehensive range of activities, to accommodate various levels of skill of the residual arm. We focused on frequency, rather than quality of movement rating, as we were interested in both prosthetic limb and stump usage, which could not have been easily assessed. We used a scale of three frequency ratings, rather than five, since pilot testing showed that individuals tended to ignore the second and fourth options. Each item was scored (between 0 and 2), and the sum was divided by 54, such that individuals were rated on a scale between 0 to 1.

To validate the usage questionnaires, and to study the relationship between increased reported residual arm usage and intact hand usage, limb acceleration data were collected from a subset of 21 limb-absent individuals (8 with congenital absence) using the GeneActiv accelerometers (sample rate: 100 Hz). Participants were given two strapped sensors, which they were asked to place on their wrist (intact hand) or the proximal aspect of the upper arm (residual arm) during 2 days with a typical daily routine (mean number of hours per day ± s.e.m. = 15:08 ± 37 min). Data from one acquired participant was discarded, due to hardware malfunction. Data for each of the three movement axes was initially smoothed using a 500 ms kernel, to discard high frequency noise. To quantify the number of movements executed with each limb, the difference between maximal and minimal acceleration values were initially calculated within a sliding window of 400 ms for each movement axis separately. In each time window, movements were identified as difference in acceleration that was above a threshold of 0.2 m/s$^2$ in at least one axis, provided that this increase was preceded and followed by periods of no movement (a difference between maximal and minimal acceleration below 0.2 m/s$^2$) in all axes. To account for whole body movements, as well as differences in number of hours of recordings, a ratio between the two limb movements was used, rather than an absolute number of movements. The movement laterality ratio was chosen [(intact−residual)/(intact+residual)], as it portrays the extent of intact hand usage, given the contribution of residual arm movements.

## Scanning procedures

Task-based fMRI: participants were visually instructed to move their left/right hand (finger movements), left/right arm (elbow movements), feet (bilateral toe movements) or lips. The protocol comprised of alternating 12 s periods of movement and 'rest'. Each of the six conditions was repeated four times, in a counterbalanced order. Here we focus on results from intact/dominant hand and residual/nondominant elbow movements (see *Makin et al., 2013*) for information about phantom movements). Participants received extensive training on the degree and form of movements expected. Note that the movements were easy to execute and did not require any expertise. To confirm that appropriate movements were made at the instructed times, task performance was monitored visually both on- and off-line, using video recordings.

## MRI data acquisition

The MRI measurements were obtained using a 3 Tesla Verio scanner (Siemens, Erlangen, Germany) with a 32-channel head coil. Anatomical data were acquired using a T1-weighted magnetization prepared rapid acquisition gradient echo sequence (MPRAGE) with the parameters: TR = 2040 ms; TE = 4.7 ms; flip angle = 8°, voxel size = 1 mm isotropic resolution. Functional data based on the blood oxygenation level-dependent (BOLD) signal were acquired using a multiple gradient echo-planar T2*-weighted pulse sequence, with the parameters: TR = 2000 ms; TE = 30 ms; flip angle = 90°; imaging matrix = 64 × 64; FOV = 192 mm axial slices. 46 slices with slice thickness of 3 mm and no gap were oriented in the oblique axial plane, covering the whole cortex, with partial coverage of the cerebellum. Two sets of whole brain diffusion weighted volumes were acquired using a generalized autocalibrating partially parallel acquisitions (GRAPPA) sequence, using the following parameters: 60 directions (plus 8 vol without diffusion weighting); b = 1000 s/mm; 65 axial slices; voxel size 2 × 2 × 2 mm; TR = 9600 ms; TE = 87 ms.

## Preprocessing and statistical analysis

All imaging data were processed using FSL 5.1 (www.fmrib.ox.ac.uk/fsl). Data collected for individuals with absent right hands (6 acquired and 3 congenital participants), were mirror reversed across the mid-sagittal plane prior to all analyses so that the hemisphere corresponding to the missing hand was consistently aligned. Data collected for left-hand dominant controls (n = 7) were also flipped, in order to account for potential biases stemming from this procedure.

## Functional analysis

Functional data were analysed using FMRIB's expert analysis tool (FEAT, version 5.98). The following pre-statistics processing was applied to each individual run: motion correction using FMRIB's Linear Image Registration Tool (MCFLIRT [*Jenkinson et al., 2002*]); brain-extraction using BET (*Smith, 2002*); mean-based intensity normalization; high pass temporal filtering of 300 s; and spatial smoothing using a Gaussian kernel of FWHM (full width at half maximum) 5 mm. Time-series statistical analysis was carried out using FILM (FMRIB's Improved Linear Model) with local autocorrelation correction. Functional data were aligned to structural images (within-subject) initially using linear registration (FMRIB's Linear Image Registration Tool, FLIRT), then optimized using Boundary-Based Registration (*Greve and Fischl, 2009*). Structural images were transformed to standard MNI space using a non-linear registration tool (FNIRT), and the resulting warp fields applied to the functional statistical summary images.

To compute task-based statistical parametric maps, we applied a voxel-based general linear model (GLM), as implemented in FEAT. The block design paradigm was convolved with a gamma function (*Friston et al., 1998*), and its temporal derivative was used to model the activation time-course at the individual level. For our main comparisons, contrasts between the intact/dominant hand and the residual/nondominant arm conditions were defined against the feet condition. Further contrasts were defined between intact/dominant arm (in the congenital group) and nondominant/missing hand (in the acquired group) vs feet movements. Group level analysis of spatial maps was carried out using FMRIB's Local Analysis of Mixed Effects (FLAME). Intact hand movements in the 1-handed groups was compared with dominant hand movements in controls, and residual arm movements were compared with non-dominant arm movements. Data from participants with above elbow deprivation (1 congenital and 7 acquired participants) were excluded from the analysis of activity during arm movements. The cross-subject GLM included planned comparisons between each of the groups against the other two, while accounting for the unbalanced comparison. Z (Gaussianised T/F) statistic images were thresholded using clusters determined by Z>2 and a family-wise-error corrected cluster significance threshold of p<0.05 was applied to the suprathreshold clusters.

To visualise the human homunculus, activation maps during feet, hands, arms and lip movements in the control group were thresholded at Z>3, each map was mirror flipped and maps representing each of the four body parts were averaged, such that a symmetrical representation of each body part was achieved. For presentation purposes, statistical parametric activation maps were projected on the inflated surface of a representative participant's cortex, using FreeSurfer.

## ROI analysis

We followed the same ROI selection and thresholding procedures as previously reported in *Makin et al. (2013)*. Briefly, it has been demonstrated that movements of a phantom limb elicit both central and peripheral motor signals that are different from those found during imagined movement (*Reilly et al., 2006*; *Raffin et al., 2012*). Since the acquired, but not the congenital participants, displayed group activation in the sensorimotor hand knob during phantom hand movements (*Makin et al., 2013*), the congenital group was excluded from the ROI definition. The 'deprived cortex' ROI was therefore defined using the conjunction of missing/non-dominant hand movements (compared to feet movements) in the acquired and control groups only. A pre-determined threshold of Z>7 was chosen, yielding a single cluster, centred on the hand knob of the central sulcus, contralateral to the missing hand. A second cluster in lobule V of the cerebellum, ipsilateral to the nondominant/phantom hand was defined separately. For within-participant comparisons, a homologous ROI for the intact hand was defined using the conjunction between controls and all 1-handed participants (using a pre-determined threshold of Z>8), yielding a comparable cluster contralateral to the intact hand. GLM parameter estimate values (β) of the low level statistical parametric maps of the contrasts described above were extracted from all voxels underlying the pre-determined ROIs and then averaged for each participant.

## White matter analysis

Voxelwise statistical analysis of the FA data were carried out using Tract-Based Spatial Statistics (TBSS [*Smith et al., 2006*]). TBSS projects all subjects' FA data onto a mean FA tract skeleton, before applying voxelwise cross-subject statistics. Diffusion data were initially corrected for eddy-currents and head motion using affine registration to the average of the non-diffusion-weighted volumes. A diffusion tensor model was fitted at every voxel to derive FA maps, which were non-linearly registered to group-specific

templates created from 11 participants of each group (FA thresholds > 0.2). The resulting FA images were temporally concatenated into a single 4D file and averaged to create a mean 'skeleton', representing the centre of all white matter tracts onto which participant-specific FA values were projected. Tract-based spatial statistics was performed voxel-wise, using a GLM and permutation-based non-parametric testing. The GLM included either planned comparisons between the three groups, or the beta values extracted from the deprived cortex ROI during intact hand movements for each of the individuals with hand absence. An age regressor of no interest was also included for the correlation analysis. Clusters were formed at t > 2 and tested for significance at p<0.05, corrected for multiple comparisons across space. For presentation purposes, the FA clusters were dilated to fill the corresponding tracts.

### Statistical analysis of limb-use and ROI-based FMRI data

For analysis of limb-use data, independent-samples two-tailed *t*-tests were used to compare congenital and acquired groups.

For ROI-based analysis of fMRI activation, intact hand movements in the 1-handed groups was compared with dominant hand movements in controls, and residual arm movements were compared with non-dominant arm movements. Data from participants with above elbow deprivation were excluded from the analysis of activity during arm movements. Between-group effects were initially statistically compared using a mixed-level ANOVA. For each limb, two comparisons were planned based on our behavioural results. Therefore, for follow-up unpaired *t*-tests of these planned comparisons, a p-value of p≤0.025 was used, to account for multiple comparisons. Other within/between-group effects were statistically compared using paired/independent-samples two-tailed *t*-tests (α < 0.05). Where significant departure from normality was found (based on the Shapiro-Wilk test), nonparametric tests were used (Man-Whitney or Wilcoxon, as appropriate).

To test for correlations between questionnaire, acceleration data, and fMRI activation, two-tailed Pearson tests were used. To exclude the involvement of various confounds from the correlation analysis (e.g., level of amputation, age at deprivation), post-hoc one-tailed partial correlation analysis was performed. To exclude the involvement of various confounds (e.g., phantom pain) from the group comparisons, univariate GLM comparisons were carried, with the confounding measurement as a covariate. To account for the potential contribution of phantom pain on the BOLD differences, chronic levels of phantom pain were assessed in each 1-handed individual. Chronic pain magnitude was calculated by dividing pain intensity (0: 'no pain'—10: 'worst pain imaginable') by frequency (1—'all the time', 2 —'daily', 3—'weekly', 4—'several times per month' and 5—'once or less per month'). This measure therefore reflects the chronic aspect of the pain as it combines both frequency and intensity, as used previously (*Draganski et al., 2006*; *Makin et al., 2013*). Comparison between correlations was assessed using a two-tailed Fisher r-to-z transformation. Statistical analysis was carried with SPSS version 18.

## Acknowledgements

Supporting bodies: Royal Society; Marie Curie Actions; European Community's Seventh Framework Programme FP7/2007-2013 under grant agreement number PITN-GA-2008-290011; Wellcome Trust; National Institute for Health Research (NIHR) Oxford Biomedical Research Centre based at Oxford University Hospitals NHS Trust and University of Oxford; MRC. We thank Nancy Rawlins for help with accelerometry data collection. We thank Scott Frey and Karen Reilly for helpful comments on a preliminary version of the manuscript. We thank Opcare for help with volunteer recruitment, and our participants for their contribution to the study.

## Additional information

### Funding

| Funder | Grant reference number | Author |
| --- | --- | --- |
| Royal Society | | Tamar R Makin |
| Marie Curie Actions | | Tamar R Makin |
| Wellcome Trust | 090955/Z/09/Z and 083259/Z/07/Z | Heidi Johansen-Berg, Irene Tracey |

| Funder | Grant reference number | Author |
|---|---|---|
| NIHR Oxford Biomedical Research centre | | Heidi Johansen-Berg, Irene Tracey |
| Medical Research Council | G0700399 | Heidi Johansen-Berg, Irene Tracey |
| European Commission | PITN-GA-2008-290011 | Heidi Johansen-Berg |

The funders had no role in study design, data collection and interpretation, or the decision to submit the work for publication.

## Author contributions

TRM, Conception and design, Acquisition of data, Analysis and interpretation of data, Drafting or revising the article; AOC, JS, DHS, Acquisition of data, Drafting or revising the article; AH, Analysis and interpretation of data, Drafting or revising the article; IT, HJ-B, Conception and design, Analysis and interpretation of data, Drafting or revising the article

## Ethics

Human subjects: Informed consent and consent to publish was obtained in accordance with ethical standards set out by the Declaration of Helsinki (1964) and with procedures approved by the NHS (REC ref: 10/H0707/29).

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
