## [Decision Letter]

Thank you for sending your work entitled “Deprivation-related and use-dependent plasticity go hand in hand” for consideration at *eLife*. Your article has been favorably evaluated by a Senior editor and 2 reviewers, one of whom, Ranulfo Romo, is a member of our Board of Reviewing Editors.

The Reviewing editor and the other reviewer discussed their comments before we reached this decision, and the Reviewing editor has assembled the following comments to help you prepare a revised submission. There are several issues that need to be addressed before the paper can be considered for publication in *eLife*:

Reviewer #1: Although this reviewer finds your paper acceptable as is, he/she requires the authors to discuss other potential alternatives in plasticity changes induced in amputees outside the input (somatosensory) output cortex (motor cortex). Also, this reviewer calls for a more precise definition of the term “sensorimotor cortex”.

Reviewer #2: This reviewer requests clarifications on data representations on Figures 2 and 3, and removing Figure 5. These actions would require reorganizing the main Results and Discussion section.

The issues to address are as follows:

1) I have always been surprised that such changes are restricted to the input-output cortex. The deprived sensory (S1: areas 3b, 3a, 1 and 2) side must alter a large neural network, including the somatosensory areas of the posterior parietal lobe, which are not seen through the imaging methodology used here. On other hand, the frontal motor areas that send massive inputs to M1 should be also altered in their patterns due to limb usage strategies. Again, nothing is seen here. In fact, I think what it might be more important in brain plasticity is adaptation changes outside the sensorimotor cortex. In fact, we know that action planning occurs outside sensorimotor areas. Perhaps the author should discuss this possibility in the discussion section as a target for neurorehabilitation, and that brain adaptive changes are not restricted to the sensorimotor cortex.

2) Can the authors show whether plasticity changes are more important in the somatosensory side than in the motor side or vice versa? I frankly do not like the term sensorimotor cortex. Neurophysiological studies clearly refer to somatosensory cortex and motor cortex.

3) Figure 2. The data show multiple representations from normals as lines, and amputee data as colored in Z-values. I think. I actually cannot determine with any confidence whether the two amputee groups are being independently plotted in Figure 2 am having to look at the figure far too long to merit this level of confusion. Can Figure 2 be clarified? Specifically, can it be clarified where the acquired and the congenital plots are?

4) Figure 3, and ROI analysis in general. It is unclear exactly what is meant by ROI by the authors. Is this an area change? Or, is the area predetermined and the percent signal change is being plotted? A title for the plot that was descriptive of the metric would be useful. If appropriate, perhaps “fMRI activation percent change in ROI”.

5) Figure 5. Inset b is weak. The statistical test used, a Pearson correlation, is already susceptible to outliers, and visual inspection of the graph shows outliers. Please remove this figure, accompanying results, and discussion from the manuscript. Such a finding would be important, but it should also be robust.

---

## [Author Response]

*1) I have always been surprised that such changes are restricted to the input-output cortex. The deprived sensory (S1: areas 3b, 3a, 1 and 2) side must alter a large neural network, including the somatosensory areas of the posterior parietal lobe, which are not seen through the imaging methodology used here. On other hand, the frontal motor areas that send massive inputs to M1 should be also altered in their patterns due to limb usage strategies. Again, nothing is seen here. In fact, I think what it might be more important in brain plasticity is adaptation changes outside the sensorimotor cortex. In fact, we know that action planning occurs outside sensorimotor areas. Perhaps the author should discuss this possibility in the discussion section as a target for neurorehabilitation, and that brain adaptive changes are not restricted to the sensorimotor cortex*.

We completely agree that adaptive brain plasticity should be studied across the entire brain, as it is likely to rely on a network of brain areas both within and beyond the sensorimotor system, supporting hand-object interactions. For this reason, our first imaging results (Figure 2) are based on whole-brain contrasts, at a relatively liberal statistical threshold (initial cluster forming threshold Z>2, family wise error corrected cluster extent threshold p<0.05). Despite this fact, the only significant cluster resulting from the two contrasts was centred on the hand knob of the central sulcus. This point is now further emphasised in the Results section of the revised manuscript.

We have also modified Figure 2 to make this result clearer (see response to comment 3 below).

An alternative, hypothesis-driven, approach to explore localised changes associated with adaptive plasticity is using ROIs. As the main purpose of our study was to identify changes in representation in the deprived cortex, and their relationship with adaptive behaviour, we focused our ROI analysis on the deprived cortex and deprived cerebellum. However, this does not rule out the possibility that similar group differences might have been revealed in other brain areas. Indeed, when studying the contrast maps shown in Figure 2, before applying correction for multiple comparisons (i.e., simply after thresholding at Z>2, uncorrected, Author response image 1), effects are found in other areas. As shown below, it appears that during intact hand movements, activation in acquired amputees is increased (compared to the other groups) not only in the deprived sensorimotor cortex and the corresponding cerebellum, but also in other areas ipsilateral to the moving hand, both within (SMA, posterior putamen) and beyond the sensorimotor system (middle and posterior insula). During residual arm movements, increased activation in the congenital group is potentially more prominent in visual areas. Given that the results presented below are highly susceptible to Type II errors, these are inappropriate to include in the paper. Nevertheless, in the revised manuscript we discuss such potential results, and highlight the importance of future research into the relationship between adaptive behaviour and activity in other brain areas that originally enabled sensorimotor control of the missing hand.Author response image 1.Group contrast maps (uncorrected for multiple comparisons) between activation patterns of intact/dominant hand (top) and residual/nondominant arm movements (bottom), as carried in Figure 2 of the main text. Thresholded clusters (Z>2, uncorrected) are presented on axial slices.

*2) Can the authors show whether plasticity changes are more important in the somatosensory side than in the motor side or vice versa? I frankly do not like the term sensorimotor cortex. Neurophysiological studies clearly refer to somatosensory cortex and motor cortex*.

We agree that the term sensorimotor is ambiguous, and that the primary somatosensory cortex (SI) could have distinct roles from the primary motor cortex (M1) in the results we present. Indeed, it has previously been demonstrated in monkeys that plasticity in SI, following arm deafferentation (using lesions to the dorsal horn), doesn't necessarily result in equivalent topographic reorganisation in M1, but rather more subtle changes in movement representations in M1 (Kambi et al. 2011). It’s therefore likely that the over-representation of the intact hand of the acquired amputees would primarily result from reorganisation in SI, whereas reorganisation of the representation of the residual arm, requiring more pronounced changes in motor control, may result in further reorganisation in M1. Indeed, when carefully studying the clusters resulting from the two whole-brain contrast maps presented in Figure 2, we find that the peak voxels are positioned within the posterior bank of the central sulcus for the contrast involving intact hand movements, and both anterior and posterior to the central sulcus for the contrast involving residual arm movements. This information is now mentioned in the Results section of the revised manuscript.

Regretfully, we believe we are not able to make any conclusive statements on this important issue. This is due to the standard spatial resolution used in this study: As described in our methods, our initial functional voxels were of 27mm (3x3x3mm), and were further smoothed (using a 5mm FWHM), resulting in overlapping sampling of SI and M1. Furthermore, to achieve group comparisons (as carried in both the whole-brain contrasts and the ROI analysis we used), individual functional maps were coregistered to a standard template, using both linear and non-linear transformations, resulting in potential further distortions and inconsistencies in alignment of the central sulcus across participants, thus making it impossible to tease apart SI from M1 with confidence. To demonstrate this point, we divided the deprived cortex ROI into two sub sections – SI and M1 (while excluding all voxels along the central sulcus, see below), and calculated mean beta values within each sub-region for the two contrasts of interest (hand and arm movements vs. feet movements). When comparing the beta values obtained from each of the sub-ROIs across participants, we found highly significant (p<0.0001) correlations between M1 and S1 values (r(49)=0.89 and 0.77 for the intact hand and residual arm contrasts, respectively; see below, Author response image 2), suggesting that these two ROIs are not independent. Indeed, the comparisons between limbless groups reported in the paper were replicated in both sub-ROIs. For this reason, we decided to continue using the term ‘sensorimotor’. Given the issues mentioned, we do not believe that these analyses add any useful information and therefore do not propose to include them in the revised manuscript (though would be willing to do so if requested). This issue is now addressed in the Results section of the revised manuscript.Author response image 2.Separate analysis of M1 and SI. (a) To explore the contributions of SI and M1 to our ROI analysis, we constructed an expanded ROI (by thresholding the conjunction maps for phantom and nondominant hand movements, used to identify the deprived cortex, at Z>6 (cf. Z>7 used for the original ROI). This expansion was necessary to obtain a reasonable sampling of voxels on each side of the central sulcus). We considered voxels anterior to the central sulcus as belonging to M1, and those posterior as belonging to S1. Voxels along the central sulcus itself were excluded as it is not possible to know whether these should be assigned to M1 or to S1. The central sulcus exclusion zone was defined based on an anatomical landmark (approximately 2-4 mm wide, extending along the central sulcus), derived from the MNI 152 template brain. Note that due to these procedures, our original deprived cortex ROI only partially overlaps with these sub-regions. We then extracted individual participants’ fMRI activation values (β) from each of the sub-regions during intact/dominant hand (left) and residual/nondominant arm (right) movement execution. As seen in (b), beta values across participants showed similar patterns in SI and M1. When comparing the averaged fMRI activation values across groups (c-d), we found highly significant differences between the two limbless populations in both sub-regions. Asterisks denote significance levels of *p<0.05; **p<0.01.

*3)*
Figure 2*. The data show multiple representations from normals as lines, and amputee data as colored in Z-values. I think. I actually cannot determine with any confidence whether the two amputee groups are being independently plotted in*
Figure 2
*am having to look at the figure far too long to merit this level of confusion. Can*
Figure 2
*be clarified? Specifically, can it be clarified where the acquired and the congenital plots are*?

We apologise for this lack of clarity. As the reviewer correctly inferred, representation of the limbs and lips, as found in the control group, was denoted in lines. The group comparisons between each of the two limbless populations (as well as the controls) were shown in coloured clusters representing z-values. The two contrasts shown in this figure were constructed based on the patterns of adaptive limb usage, found in Figure 1: for the congenital population, we contrasted residual arm movements against the two other groups; for the acquired population, we contrasted intact hand movements against the two other groups. Both contrasts resulted in a similar map – a single cluster centred on the hand knob of the central sulcus. In the revised manuscript we’ve adjusted the figure, such that the borders of the different body parts in controls are plotted separately. The overlap between the controls’ “homunculus” and the two independent contrast maps is shown in separate inserts. We hope that this new presentation will also help clarify that the activation maps shown in the figure resulted from a whole-brain comparison and were not restricted by any masks.

*4)*
Figure 3*, and ROI analysis in general. It is unclear exactly what is meant by ROI by the authors. Is this an area change? Or, is the area predetermined and the percent signal change is being plotted? A title for the plot that was descriptive of the metric would be useful. If appropriate, perhaps “fMRI activation percent change in ROI”*.

We apologise for this confusion. As described in the Methods section, the ROI consists of a cluster of voxels selectively activated by the conjunction of phantom and nondominant hand movements (compared to feet movements) in acquired amputees and controls. The threshold of this cluster has been predetermined, based on our previous publication (17), which focused on activity relating to the phantom. In the current submission, that same ROI is used to interrogate activity relating to other body parts. The values used in the ROI analysis are the averaged beta values (across all the voxels in this predetermined cluster), quantifying fMRI activation during intact/dominant hand or residual/nondominant arm movements for individual participants. We detail this information in the Methods section of the revised manuscript.

We also adjusted the labels in Figure 3 as suggested. Finally, we added inserts of the ROI boundaries and its overlap with activation patterns from Figure 2 in the Figure. We hope this will help elucidate the ROI analysis.

*5)*
Figure 5*. Inset b is weak. The statistical test used, a Pearson correlation, is already susceptible to outliers, and visual inspection of the graph shows outliers. Please remove this figure, accompanying results, and discussion from the manuscript. Such a finding would be important, but it should also be robust*.

We take the reviewers’ point that the latter correlation is weak and is partly driven by an outlier. However, the point we were trying to make is that the correlation in the limbless population differed from that found in controls. Indeed, we find highly significant difference between the two correlations (p=0.001) also when comparing the Spearman’s Rho values of the two correlations (using Fisher’s r to z transformation). As this figure was originally designed to convey a secondary issue, we removed it from the revised manuscript. However, should the editors feel that given the clarification above this result is informative, we will be happy to mention it in the Results.